# Evaluating Different TiO_2_ Nanoflower-Based Composites for Humidity Detection

**DOI:** 10.3390/s22155794

**Published:** 2022-08-03

**Authors:** Musa Mohamed Zahidi, Mohamad Hafiz Mamat, Mohd Firdaus Malek, Muhamad Kamil Yaakob, Mohd Khairul Ahmad, Suriani Abu Bakar, Azmi Mohamed, A Shamsul Rahimi A Subki, Mohamad Rusop Mahmood

**Affiliations:** 1NANO-ElecTronic Centre (NET), School of Electrical Engineering, College of Engineering, Universiti Teknologi MARA, Shah Alam 40450, Selangor, Malaysia; musa_zahidi59@uitm.edu.my; 2Centre for Electrical Engineering Studies, Permatang Pauh Campus, Universiti Teknologi MARA Cawangan Pulau Pinang, Permatang Pauh 13500, Pulau Pinang, Malaysia; 3NANO-SciTech Lab (NST), Centre for Functional Materials and Nanotechnology, Institute of Science (IOS), Universiti Teknologi MARA, Shah Alam 40450, Selangor, Malaysia; mfmalek07@uitm.edu.my (M.F.M.); muhamadkamil@uitm.edu.my (M.K.Y.); rusop@uitm.edu.my (M.R.M.); 4Faculty of Applied Sciences, Universiti Teknologi MARA, Shah Alam 40450, Selangor, Malaysia; 5Microelectronic and Nanotechnology–Shamsuddin Research Centre, Faculty of Electrical and Electronic Engineering, Universiti Tun Hussein Onn Malaysia, Batu Pahat 86400, Johor, Malaysia; akhairul@uthm.edu.my; 6Nanotechnology Research Centre, Faculty of Science and Mathematics, Universiti Pendidikan Sultan Idris, Tanjung Malim 35900, Perak, Malaysia; suriani@fsmt.upsi.edu.my (S.A.B.); azmi.mohamed@fsmt.upsi.edu.my (A.M.); 7Faculty of Electrical and Electronic Engineering Technology, Universiti Teknikal Malaysia Melaka, Hang Tuah Jaya, Durian Tunggal 76100, Melaka, Malaysia; shamsulrahimi@utem.edu.my

**Keywords:** TiO_2_ nanoflower, solution immersion method, composite, humidity sensor

## Abstract

Unique three-dimensional (3D) titanium dioxide (TiO_2_) nanoflowers (TFNA) have shown great potential for humidity sensing applications, due to their large surface area-to-volume ratio and high hydrophilicity. The formation of a composite with other materials could further enhance the performance of this material. In this work, the effect of different types of composites on the performance of a TNFA-based humidity sensor was examined. NiO, ZnO, rGO, and PVDF have been explored as possible composite pairing candidates with TiO_2_ nanoflowers, which were prepared via a modified solution immersion method. The properties of the composites were examined using field emission electron spectroscopy (FESEM), X-ray diffractometry (XRD), transmission electron microscopy (TEM), X-ray photoelectron spectroscopy (XPS), current-voltage (I-V) analysis, Hall effect measurement, and contact angle measurement. The performance of the humidity sensor was assessed using a humidity sensor measurement system inside a humidity-controlled chamber. Based on the result, the combination of TiO_2_ with rGO produced the highest sensor response at 39,590%. The achievement is attributed to the increase in the electrical conductivity, hydrophilicity, and specific surface area of the composite.

## 1. Introduction

Nanostructured metal oxides are fast becoming one of the most important materials for sensing applications. In particular, for chemical sensing whereby the sensing mechanism often depends on the adsorption and desorption process, the large surface area of these materials is advantageous to ensure a sensitive and accurate measurement. A humidity sensor is one type of chemical sensor that fits into this category. The detection mechanism of a resistive-type humidity sensor is through changes in the electrical resistance when water molecules are attached to or detached from the surface of the sensing material. Titanium dioxide (TiO_2_) stands out as one of the preferred materials for humidity detection. With its high hydrophilicity [1], chemical stability, and low cost, this compound has emerged as a perfect building block for a humidity sensor. The presence of Ti^3+^ defect sites is also known to promote the efficient adsorption of water molecules [2,3]. In addition, TiO_2_ can also be made into myriad types of nanostructures with a large surface area, such as nanorods, nanotubes, nanoballs, and nanoflowers [4,5,6,7].

A material with a nano-sized structural morphology has shown remarkable performance when employed in humidity sensor applications. Jyothilal et al. [8] reported on the performance of a humidity sensor prepared using slanted TiO_2_ nanorods. The improved sensitivity is attributed to the increased surface area for water vapour interaction and condensation. Li et al. also produced an ultrasensitive humidity sensor using TiO_2_ nanowires that are connected to Ti_3_C_2_, utilizing the cross-linked structure to improve the conductivity [9]. Ultrafine TiO_2_ nanoparticles (4 to 15 nm) synthesized using the aero-gel method resulted in an excellent sensing performance due to its porous nature and large interlayer distance [10]. Meanwhile, Wang et al. [11] reported that an urchin-like CuO nanostructure produced a remarkable humidity sensing response due to the capillary condensation process, facilitated by the gap existing between the spine structures. Similar results have also been reported for devices prepared using hollow balls [12], nanosheets [13], nanofibers [14], nanoflakes [15], nanotubes [16], and other structures.

The performance of a humidity sensor could also be improved by mixing TiO_2_ with other materials to produce composites. Si et al. [17] added NaNbO_3_ to TiO_2_ and reported an increase in the humidity sensing performance in terms of sensitivity, rapid response/recovery time, and negligible hysteresis. They argued that the formation of a heterojunction in the TiO_2_/NaNbO_3_ composite helps induce this superior performance. Similarly, Park et al. [18] reported that this heterojunction could improve the sensing ability of the sensor by decreasing the electron current route. The potential barrier increases at the humidity condition, because water vapour adsorbed on the TiO_2_/rGO behaves as an electron acceptor. The addition of a polymer is also an interesting strategy, as it is said to have a higher polarization ability due to the contrasting chemical and physical structure. Sasikumar et al. [19] reported that a PANI/TiO_2_ composite-based humidity sensor exhibited a higher sensitivity with a faster response/recovery time. Another possible advantage of a composite is the increase in the overall surface area of the composite by manipulating the nanostructured nature of the pairing material. Araujo et al. [20] utilized ZnO nanorods with a large surface area to deliver a higher sensitivity humidity sensor. Meanwhile, a number of researchers [21] also explored the benefit of having increased Ti^3+^ or oxygen vacancy defects induced by the formation of the composite on the performance of the humidity sensor. They reported on the enhanced hydrophilicity of the sensing material due to the strong electrostatic field around the defect sites.

Previously, our group managed to produce novel TiO_2_ flower-like nanorod arrays (TFNA) on glass substrates using a facile solution immersion method [22]. Our method is unique because it does not require the use of an autoclave. Instead, we utilize a Schott bottle and a home-made clamp to produce TFNA with a high surface area. In an effort to further improve the sensitivity value of the humidity sensor, in this work, we experimented with TiO_2_/NiO, TiO_2_/rGO, TiO_2_/ZnO, and TiO_2_/PVDF composites with different structures, as illustrated in Figure 1. Herein, the comparison between these composites is presented and discussed. Literature concerning the incorporation of different materials into the TiO_2_ nanoflower structure for resistive-type humidity sensor applications is still relatively scarce. This work will explore the possibility of these composites in terms of their humidity detection capability.

## 2. Materials and Methods

The procedure started with the preparation of a seed layer by depositing a thin layer of TiO_2_ on a soda-lime glass substrate via radio frequency (RF) sputtering (SNTEK). Pure TiO_2_ (99.99% purity) was utilized as the target at an RF power of 200 W with 20 and 5 sccm flowrate of argon and oxygen gas, respectively. An adjustable throttle valve automatically controlled the chamber pressure so that it was maintained at 5 mTorr throughout the deposition process. The deposition duration of 6 h yielded an approximately 370 nm thick TiO_2_ film. For the preparation of the growth solution, hydrochloric acid (HCl) and ultra-pure water were combined in a Schott bottle at the ratio of 1:1 and stirred for 10 min. Next, 0.07 M of titanium butoxide (97% purity, Sigma-Aldrich, St. Louis, MO, USA) was added to the mixture and stirred for an additional 50 min. A previously prepared TiO_2_ seed-layer-coated glass substrate was carefully positioned inside the Schott bottle with the seed layer facing in the upside direction. The bottle was tightly sealed with a heat-resistance cap and an improvised clamp before being placed inside the oven for 4 h at a temperature of 150 °C. Following the immersion process, the samples were washed with ultra-pure water and calcined at 500 °C for 1 h. This formed the base TFNA thin films.

The TiO_2_/NiO composite was produced by adding a NiO layer on TFNA using a method similar to that described by Parimon et al. [23]. A glass substrate with a TFNA layer was prepared and inserted inside a Schott bottle containing 0.2 M nickel (II) nitrate hexahydrate (Ni(NO_3_)_2_·6H_2_O; 97% purity; Friendemann Schmidt), 0.2 M hexamethylenetetramine (C_6_H_12_N_4_; 99% purity; Sigma-Aldrich), and ultra-pure water, which was subsequently immersed in a water bath at 75 °C for 3 h. The sample was then retrieved and rinsed with ultra-pure water.

For the fabrication of the TiO_2_/ZnO composite, the TFNA was placed in a Schott bottle containing a solution prepared by the method described by Ismail et al. [24]. The solution consists of zinc nitrate hexahydrate ((Zn(NO_3_)_2_·6H_2_O; 98.5% purity; Friendemann Schmidt), 0.001 M iron (III) nitrate nanohydrate (Fe(NO_3_)_3_·9H_2_O; 98% purity; Merck), and 0.1 M hexamethylenetetramine (C_6_H_12_N_4_; 99% purity; Sigma-Aldrich). The bottle was submerged in 95 °C water bath for 1 h. The substrate was then removed and rinsed with DI water.

The TiO_2_/rGO composite was produced by adding a layer of rGO on the TFNA via the drop-casting method. The rGO powder was dispersed in ethylene glycol monoethyl ether at a concentration of 0.03 mg/mL. The solution was rigorously stirred using a magnetic stirrer and ultrasonic bath to obtain a uniformly dispersed solution. The solution was drop-casted on the TFNA using a micropipette and dried on a hot plate stirrer to remove the dispersant. 

Finally, in the case of the TiO_2_/PVDF composite, the PVDF layer was prepared according to the step introduced elsewhere [25]. PVDF powder ((CH_2_CF_2_)_n_; Sigma-Aldrich) was dissolved in N-dimethylformamide (DMF, HCON(CH_3_)_2_, Sigma-Aldrich) and stirred rigorously for 1 h. The result was a clear solution that was then spin-coated on the TFNA layer. The sample was then heated in an oven at 100 °C to remove the residual solvent. 

A thermal evaporator (ULVAC) was then employed to deposit a silver (Ag) metal contact pattern for electrical properties measurement using a metal mask. Field-emission scanning electron microscopy (FESEM, JEOL JSM-7600F) was used to examine the morphology of the prepared materials. Micro-Raman spectroscopy (Horiba Jobin Yvon-79 DU420A-OE-325, 514 nm Ar laser), X-ray diffraction (XRD, PANalytical X’pert PRO), and high-resolution transmission electron microscopy (HRTEM, FEI TECNAI G2 20 S-TWIN) were used to characterize the morphology and the structural properties of the samples. The chemical states of the materials were analyzed using X-ray photoelectron spectroscopy (XPS, Thermo Scientific Nexsa G2). The electrical properties were examined using a two-point-probe current–voltage (I-V) measurement apparatus (Advantest R6243) and a Hall effect probing system (ezHEMS, Nanomagnetics Instrument). For the humidity sensing characteristics, the measurement was performed in a humidity-controlled chamber (ESPEC-SH261) connected to a current–voltage–time (I–V–t) measurement system (Keithley 2400). The schematic diagram of the measurement system is illustrated in Figure 2.

## 3. Results and Discussion

### 3.1. Structural Properties

The FESEM images of all the prepared materials are shown in Figure 3a–e. The pure TiO_2_ sample in Figure 3a shows the TiO_2_ nanoflower (TFNA) structure that consisted of nanorods having an average diameter of around 150 nm. The average flower diameter is calculated to be around 4 µm. Meanwhile, the TiO_2_/NiO sample shows the presence of the NiO spherical structure, referred to as NiO nanoballs. Closer inspection reveals that the structure consists of a porous sheet. The diameter of the nanoballs was determined to be around 2 µm. Large voids are observed between the nanoball structures. 

The TiO_2_/ZnO composite exhibits a mixture of flower-like nanorod arrays of TiO_2_ and dispersed nanorods of ZnO. The ZnO nanorods can be distinguished from the TiO_2_ nanorods by their hexagonal shape, whilst TiO_2_ nanorods have more of a square shape. The average diameter of the ZnO nanorods is calculated to be 727 nm. Meanwhile, only the TFNA structure is observed for the TiO2/rGO composite because it is difficult to see the graphene structure due to its small size. On the other hand, the PVDF is clearly visible in the TiO_2_/PVDF composite, manifested by the presence of blocks of polymer structure among the TFNA. EDS analysis result is shown in Figure 4a–d. The presence of composite materials is evidenced by the existence of element peaks in the EDS spectra. 

The XRD spectra of the samples are presented in Figure 5a. A reference to JCPDS card No. 01-072-1148 indicates that all the samples show the presence of rutile TiO_2_. The most prominent peak was observed at 2*θ* of 27°, which corresponds to the (110) TiO_2_ plane. The pristine TiO_2_ sample exhibits the highest peak at this point because the thin film consisted of only TiO_2_ nanoflowers. The TiO_2_/NiO composite sample also shows peaks at 42°, indicating the NiO (200) plane. The interplanar spacing, $d_{hkl}$, and lattice constant, a, taken at the TiO_2_ (110) plane were calculated using the following equations:(1)$$d_{hkl}=\frac{n\lambda }{2sin\theta }$$
(2)$$\frac{1}{d_{hkl}{}^{2}}=\frac{h^{2}+k^{2}}{a^{2}}+\frac{l^{2}}{c^{2}}$$
where n denotes value for order of diffraction (normally 1), λ denotes X-ray wavelength of 1.54 Å, θ refers to the diffraction angle, and h,k, l is the Miller indices of the planes and the result is tabulated in Table 1. The values are found to be comparable with the results from other works [26,27,28]. Meanwhile, the crystallite size and microstrain value of the samples were ascertained using the Williamson-Hall method based on the following equation:(3)$$\beta cos\theta =\frac{K\lambda }{D}+4\varepsilon sin\theta$$
where β is the full width at half maximum (FWHM), θ denotes the diffraction angle, K is the dimensionless shape factor for the crystal, D refers to the crystallite size, and ε is the microstrain of the crystal. By plotting βcosθ against 4εsinθ, the crystallite size could be determined from the slope while the intercept value determine the microstrain [29]. The calculated result is shown in Table 1 and Figure 5b. The pure TiO_2_ sample has the largest crystallite size at 97.2 nm followed by TiO_2_/NiO (75.4 nm), TiO_2_/PVDF (65.5 nm), TiO_2_/ZnO (50.5 nm), and TiO_2_/rGO (38.9 nm). The addition of a foreign compound often affects the crystal growth of TiO_2_, resulting in a smaller crystallite size. Accordingly, the microstrain value also increases in all the composites except for TiO_2_/ZnO. This might be due to the deformation, grain refinement and straining induced by foreign materials [30]. In their work, Luo et al. reported that the presence of rGO could suppress the grain growth of TiO_2_, consequently increasing the oxygen vacancy defect at the surface [31]. 

Raman analysis was performed on the pristine TiO_2_ and TiO_2_/rGO samples to further investigate the presence of carbon material, and the result is presented in Figure 6. Four distinct peaks have been detected at raman shift of 143, 235, 447, and 612 cm^−1^ which are referred to as $B_{1g}$, ∗, $E_{g}$, and $A_{1g}$, respectively. $B_{1g}$ is associated with vibration mode while ∗ is associated with multiple phonon scattering processes [32]. The raman mode of $E_{g}$ and $A_{1g}$ originated from O-Ti-O bending vibration and Ti-O stretching vibration, respectively [33]. The presence of these peaks suggested that the TiO_2_ crystals are in rutile phase. The TiO_2_/rGO film displays D and G bands at 1376 cm^−1^ and 1592 cm^−1^, respectively, indicating the presence of graphene in the film, and the *I_D_/I_G_* ratio was calculated to be 0.9366. The breathing mode of the k-point phonons of the *A_g1_* symmetry is associated with the D band, while the G band corresponds to the *E_g_* mode of sp^2^ domains of the rGO species [34]. Meanwhile, the pristine TiO_2_ shows no presence of the D and G bands. The *I_D_/I_G_* ratio is often considered to correlate with the number of defects in graphene [35]. The value is generally higher for rGO compared to GO, since the defects are usually higher in rGO due to the reduction inducing processes, such as chemical [36], heat [37], and plasma treatment [38]. Increased in *I_D_/I_G_* ratio also signals to reduction in sp^2^ lattice size [39]. A value between 0.9 and 1 is commonly observed for rGO [38].

### 3.2. Electrical Properties

The electrical characteristics of the prepared materials were examined using a two-point probe I-V measurement system, and the result is shown in Figure 7a,b. All samples showed a linear response, which indicates good ohmic contact with the Ag metal contact. The resistivity, ρ, of the films was computed based on the following formula.
(4)$$\rho =\frac{RA}{L}$$
where R represents the resistance, A denotes the area, and L is the distance between the metal contact. The TiO_2_/rGO showed the lowest resistivity among all composites. RGO is known to be a very conductive material due to its superior electron mobility [40]. It is reported that at room temperature, its electron mobility could reach up to 200,000 cm^2^V^−1^s^−1^ [41]. It is believed that the addition of rGO manages to reduce the overall resistivity of the composite.

The Hall effect measurement was carried out using a four-point probe, and the result is tabulated in Table 2. The carrier concentration of the TiO_2_ sample was recorded at 8.24 × 10^14^ cm^−3^, which is comparable with the result reported elsewhere [42]. It is also noteworthy that the TiO_2_/rGO sample showed the highest mobility among all the samples. As mentioned earlier, because graphene is famed for its high carrier mobility, hence, the addition of rGO to TiO_2_ resulted in the increased electron mobility of the sample. Other composites showed mobility values that were more or less comparable with that of the pure TiO_2_.

### 3.3. Humidity Sensor Performance

The variations in the current signal of the TiO_2_ composite samples as a function of the relative humidity (RH) level are depicted in Figure 8a. All samples show a steady increase in the current value as the moisture level was increased from 40 to 90%RH before reverting back to the initial value when the humidity was reduced back to 40%RH, which indicates that all samples are responding well to the changes in the humidity level. The following equation [43] was used to calculate the sensor response value:(5)$$S=\frac{R_{40}-R_{90}}{R_{90}}\times 100\%$$
where $R_{40}$ denotes the steady resistance at 40% RH, while $R_{90}$ is the resistance at 90% RH. The measurement was performed for 5 cycles, and the average value was taken. Figure 8b shows the average sensitivity values of the prepared sensors with their error bars. All samples show a high sensitivity of more than 18,000%, which indicates that an ultra-sensitive humidity sensor could be fabricated using the TiO_2_ composite films. From the calculation, the TiO_2_/rGO sample recorded the highest sensor response value at 39,590%, followed by TiO_2_/PVDF, TiO_2_/ZnO, and TiO_2_/NiO. For the TiO_2_/NiO sample, there is a slight decrease in the sensor response value when compared with the undoped sample. The humidity response is greatly influenced by the density of oxygen vacancies in the oxide materials [13,17]. The oxygen vacancies possess high affinity to water molecules and facilitate the adsorption sites for water molecules to enhance humidity sensing response in the process. The n-type material of TiO_2_ has high oxygen vacancies, whereby the p-type material of NiO has high Ni deficiencies [13,23]. In this work, the growth of NiO nanoballs on TiO_2_ nanoflower layer might be too excessive as observed in the FESEM image (Figure 3b), as the NiO nanoballs dominate the top surface of the film. This condition debilitates the oxygen vacancies’ role to capture more water molecules due to the existence of high-density Ni deficiencies. The performance degradation when having excessive NiO materials in the composite was also observed in other applications [44,45,46]. In addition, NiO is a known p-type semiconductor, which could compensate electron and reduce the carrier concentration in the process. The FESEM images also reveal the presence of large voids on the surface of the thin film. These voids might have prevented the smooth transition of electrons, affecting the sensor response of the device. For the TiO_2_/ZnO sample, it has been documented that the n-n composite heterostructure improves the sensitivity of the sensor. According to Yadav et al., ZnO and TiO_2_ have electron vacancies that can strongly attract H^+^ ions from water molecules [47]. Meanwhile, the TiO_2_/PVDF sample recorded the second highest sensor response at 29,260%. According to Mallick et al., PVDF is an inherently hydrophobic material. The addition of TiO_2_ improved the surface wettability of the TiO_2_/PVDF composite and subsequently improved its hydrophilicity, which resulted in the improved humidity sensing performance [25]. A breath detection test was also conducted for the TiO_2_ and TiO_2_/rGO samples. A human subject breathed near the sensor and the current changes were recorded for several cycles. The results are shown in Figure 8c. Both samples displayed rapid response and recovery to human breath. However, the TiO_2_/rGO sample exhibited higher current signal and response compared to the TiO_2_ sample only. A comparison of the humidity sensing performance between our device and the devices from the works of other groups is summarized in Table 3.

There are several factors that could be associated with the enhanced performance of the TiO_2_/rGO composite. It is reported that the addition of 2D GO could increase the specific surface area of a composite [31,57,58]. Graphene itself tends to be hydrophobic in nature, with its wetting characteristic being similar to that of graphite [59]. On the other hand, TiO_2_ has high hydrophilicity, owing to the presence of water dissociative sites of the Ti^3+^ defect. Adding both of these materials together yields a hydrophilic surface suitable for the adsorption of water molecules. Another possible reason behind the improvement in the sensor response is the effect of the high conductivity of rGO due to its high electron mobility [60]. On its own, TiO_2_ is a high electrical resistance material [57]. However, when coupled with rGO, the overall resistance of the composite is reduced greatly, as reported by researchers [40]. RGO could serve as an efficient electron channel because of its high electron mobility [58].

### 3.4. HRTEM and XPS Analysis of TiO_2_/rGO

To really understand the reason behind the excellent performance of the TiO_2_/rGO composite-based humidity sensor, further analysis on the sample was carried out. The HRTEM images of the TiO_2_/rGO composite are shown in Figure 9a. Both the rGO sheet and TiO_2_ nanorods are clearly visible. From the HRTEM image, the close proximity of TiO_2_ and rGO suggested that the transfer of electrons between the two materials could take place, as reported by Sun et al. [61].

The high-magnification HRTEM image in Figure 9b reveals the d spacing value of 0.32 nm, which is in line with the (110) plane. This value is consistent with the calculation made earlier using the XRD data. The selected area electron diffraction (SAED) pattern (inset Figure 9b) reveals rings of the d spacing, indicating the presence of the (110) plane, which is consistent with polycrystalline rutile TiO_2_.

The XPS survey scan spectra of the TiO_2_ and TiO_2_/rGO sample is presented in Figure 10a. Both samples showed the presence of C 1s, Ti 2p, Ti 2s, and O 1s peaks, with the TiO_2_/rGO sample showing a higher intensity C 1s peak, due to the presence of rGO on the TiO_2_ nanoflower surface. The narrow scan of Ti 2p in Figure 10b,c reveals two prominent peaks at the binding energies of 459 eV and 464 eV corresponding to Ti 2p_3/2_ and Ti 2p_1/2_, respectively, which arises from spin-orbit splitting. A shoulder peak at 460 eV is noticeable in the TiO_2_/rGO sample, which suggested the presence of Ti^3+^ species [62,63]. This species is often considered responsible for the adsorption of water molecules by providing ample active sites [64]. Figure 10d reveals the narrow scan of C 1s. The deconvolution of the spectra reveals 5 peaks that can be assigned to carbon from C=C sp^2^, C-C sp^3^, C-O, C=O, and O-C=O bonds. Finally, Figure 10e,f show the core level scan O 1s of TiO_2_ and TiO_2_/rGO, respectively. The main peak at 530 eV is attributed to the lattice oxygen. For the TiO_2_/rGO sample, a peak is also observed at 530.7 eV, which corresponds to the oxygen vacancy [65]. Both samples also displayed a peak at 532.1 eV corresponding to chemisorbed oxygen [64]. It is reported that this oxygen species could play a part in the redox reaction on the surface of the material.

### 3.5. Hydrophilicity of TiO_2_/rGO

The contact angle measurement was conducted for the rGO only thin film and TiO_2_/rGO composite thin films to examine the hydrophilicity of the samples. As observed in Figure 11a,b, the contact angle of rGO alone is 73.4°, which indicates a hydrophobic surface. On the other hand, the TiO_2_/rGO contact angle was recorded at almost 0°, suggesting that the addition of the TiO_2_ nanostructure greatly improves the hydrophilicity of the sample. It is well documented that the Ti^3+^ defect sites increase the hydrophilicity of TiO_2_ by promoting the dissociative adsorption process [66]. The high hydrophilicity is obviously beneficial for the humidity sensor application, as it increases the amount of adsorbed water molecules.

### 3.6. Humidity Detection Mechanism

The humidity detection mechanism of the TiO_2_/rGO nanocomposite sensor could be explained by the following. At a low humidity level (40% RH), free electrons at the surface of TiO_2_ react with the surrounding oxygen atoms according to the following reactions:(6)$$O_{2}+e^{-}\to O_{2}^{-}$$
(7)$$O_{2}^{-}+e^{-}\to 2O^{-}$$
(8)$$2O^{-}+e^{-}\to O^{2-}$$

This reaction traps the free electron, thus reducing the number of carriers of the sensing element and increasing the resistance. At this point, the resistance between the metal contact is very high. When the humidity increases, water reacts with the adsorbed oxygen ions, as described by the following equation.
(9)$$2H_{2}O+O_{2}^{-}\to 2H_{2}O_{2}+e^{-}$$

Free electrons would be released back through this chemisorption reaction, and the resistance would decrease.

When the humidity level is further elevated, physisorption takes place whereby another layer of water attaches to the chemisorbed water layer through hydrogen bonds. Under the presence of an electrostatic force, hydronium ion $H_{3}O^{+}$ would be formed by the following equation.
(10)$$2H_{2}O\to H_{3}O^{+}+OH^{-}$$

Hydrolysis occurs and $H_{3}O^{+}$ would subsequently be converted to $H^{+}$ ions.
(11)$$H_{3}O^{+}\to H_{2}O+H^{+}$$

This ion would contribute to the increase in the current in a phenomenon called the Grotthuss chain reaction or proton hopping. This type of conduction is called the ionic conduction. If the humidity is then further increased, water molecules would fill up the spaces between the TiO_2_ structure, driven by capillary force. From here on, $OH^{+}$ and $H^{+}$ ions can move freely, and electrolytic conduction takes over.

The addition of rGO with its high electron mobility accelerated the changes in reducing the resistance of the thin film. As shown in Figure 9a, the interconnection between rGO and the TiO_2_ nanorod structure would enable the efficient electron transfer between them. Meanwhile, the increased oxygen vacancies helped increase the electron concentration through the following reactions [31,67]:(12)$$H_{2}O\left(gas\right)\underset{}{\to }H_{2}O^{+}\left(ads\right)+e^{-}$$
(13)$$H_{2}O\left(gas\right)+O^{2-}\left(ads\right)+V_{o}^{2-}\underset{}{\to }2OH^{-}\left(ads\right)+2e^{-}$$

The humidity detection mechanism of rGO is described by the following. At a lower RH level, the initial layer of water molecules attaches to rGO via double hydrogen bonding. This strong bonding limits the hopping transfer of protons, leading to a low conductivity. Meanwhile, the second layer of water is physisorbed on the oxygenated groups of graphene through single hydrogen bonding and is more likely to move freely. Under the influence of an electrostatic field, the ionization of the water molecules takes place and a huge number of hydronium ions ($H_{3}O^{+}$) are formed. A subsequent increase in the RH saturates the active sites on the graphene surface, allowing more free movement of excess water molecules. Hydronium ions are transported freely in the water layers through the Grotthuss chain reaction ($H_{2}O+H_{3}O^{+}=H_{3}O^{+}+H_{2}O$). As a result, the resistance is drastically reduced by the rapid transmission of ions. In addition, at an increased humidity level, water molecules are able to penetrate into the interlayer of rGO, leading to the hydrolysis of the oxygenated group, which would also increase the conductivity [41].

The proposed energy band diagram at the interface between TiO_2_ and rGO is exhibited in Figure 12a–c. The band gap energy, $E_{g}$ of rutile TiO_2_ is 3.0 eV [68] while the work function, ϕ and electron affinity, χ are 4.2 [69] and 3.9 eV [70], respectively. Meanwhile, the work function, ϕ of rGO is 4.75 eV [71]. It is estimated that a Schottky junction forms between TiO_2_ and rGO when they are in contact with each other due to the appropriate difference in work function between the two materials, as shown in Figure 12a. At a low humidity level, oxygen atoms from the air are adsorbed by TiO_2_ (Figure 12b). This process lowers the number of free electrons, and the depletion region becomes wide, prohibiting the transfer of charge across the junction. At this point, the resistance value between the metal contact is quite high. When water molecules in the environment increases, H_2_O, which is a reducing agent, increases the number of electrons in TiO_2_, which would be transported from the conduction band of TiO_2_ to rGO, effectively reducing the Schottky barrier with a narrower depletion region, as shown in Figure 12c. This reduces the depletion region width, and the conductivity between the material increases dramatically, which translates to the increase in the humidity sensor response. The synergistic effect of the combination of these two types of humidity sensing mechanisms produced an improved sensor response compared to the sensor response of a pure TiO_2_ based humidity sensor.

## 4. Conclusions

Resistive-type humidity sensors were fabricated and tested using various types of TiO_2_ composites. The combination of TiO_2_ and rGO produced the best result with the highest sensor response of 39,590%, followed by TiO_2_/PVDF, TiO_2_/ZnO, and TiO_2_/NiO. The improved performance of the TiO_2_/rGO composite is associated with the enhanced conductivity and hydrophilicity of the sensing material. The introduction of rGO induced the formation of Ti^3+^ sites and oxygen vacancy, as evidenced by the XPS result. In addition, the nanocomposite is also postulated to have a higher surface area which provides more adsorption sites. Furthermore, the formation of the Schottky junction between TiO_2_ and rGO allows for the increased modulation of the sensing current before and after exposure of the water molecules.

## Figures and Tables

**Figure 1 sensors-22-05794-f001:**
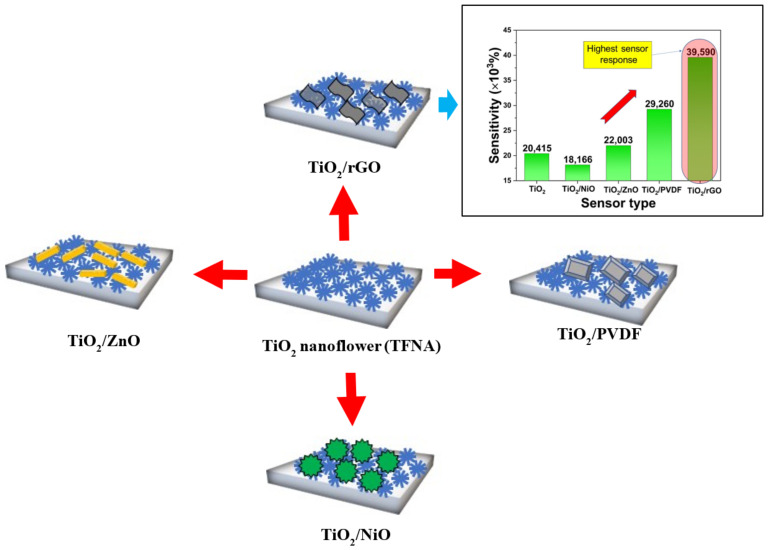
Illustration of the different composites structure proposed in this work.

**Figure 2 sensors-22-05794-f002:**
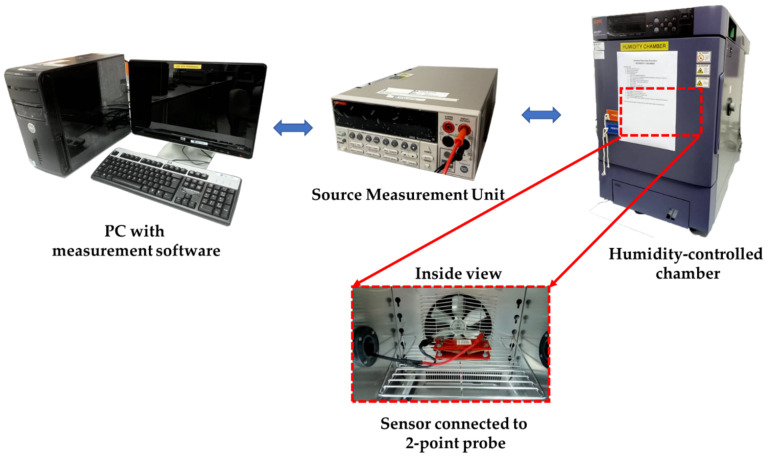
Schematic diagram of the humidity sensor measurement system.

**Figure 3 sensors-22-05794-f003:**
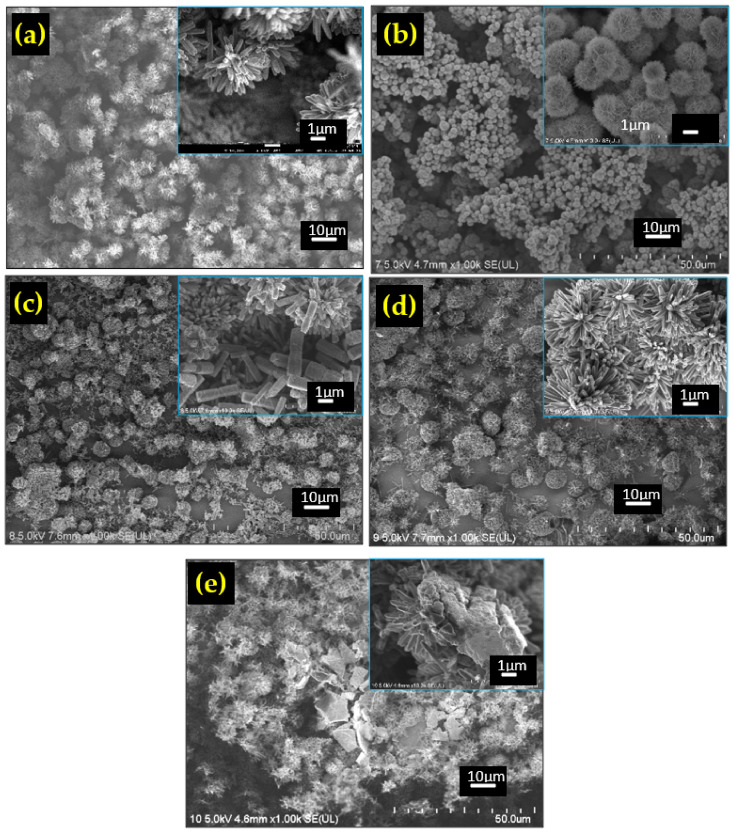
FESEM images of: (**a**) TiO_2_, (**b**) TiO_2_/NiO, (**c**) TiO_2_/ZnO, (**d**) TiO_2_/rGO, and (**e**) TiO_2_/PVDF.

**Figure 4 sensors-22-05794-f004:**
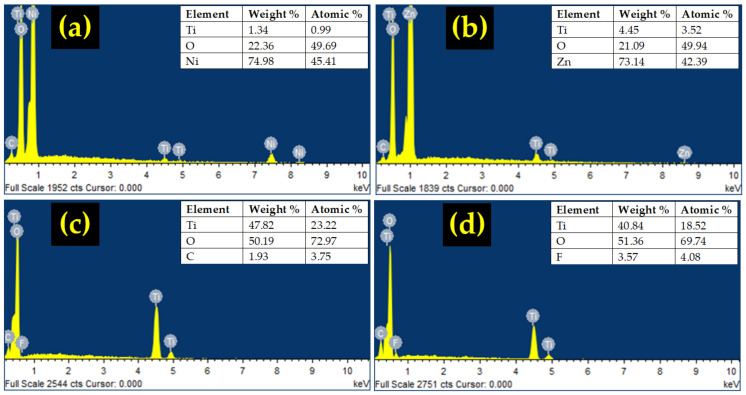
EDS analysis of: (**a**) TiO_2_/NiO, (**b**) TiO_2_/ZnO, (**c**) TiO_2_/rGO, and (**d**) TiO_2_/PVDF.

**Figure 5 sensors-22-05794-f005:**
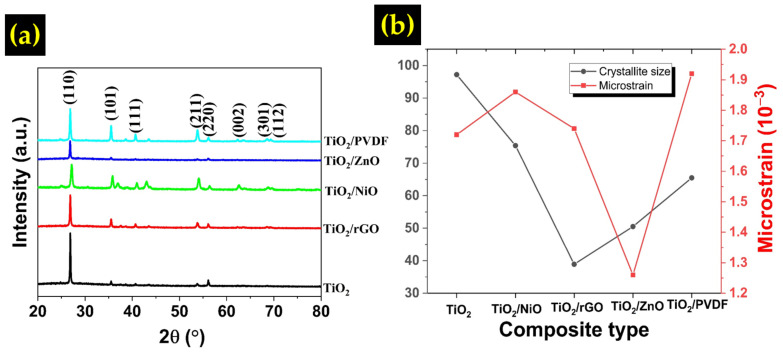
(**a**) XRD patterns of different TiO_2_ composites, (**b**) Crystallite size and microstrain of all samples.

**Figure 6 sensors-22-05794-f006:**
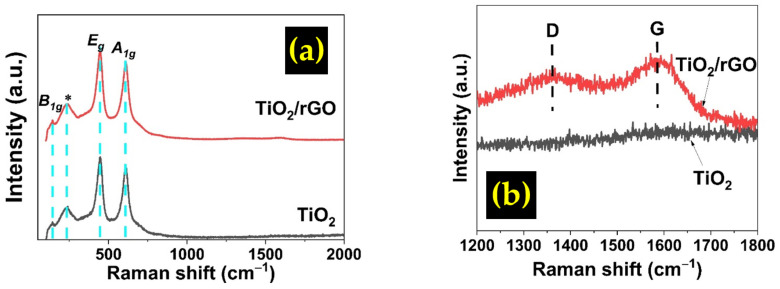
Raman spectra of TiO_2_ and TiO_2_/rGO samples at the Raman shift range of: (**a**) 0–2000 (cm^−1^) and (**b**) 1200–1800 (cm^−1^).

**Figure 7 sensors-22-05794-f007:**
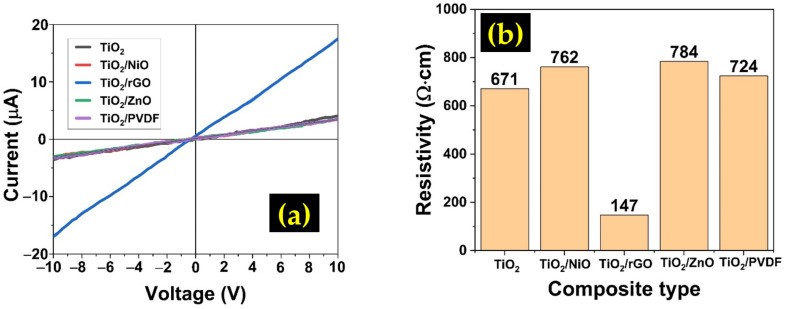
(**a**) I-V measurement result of different types of TiO_2_ composites, (**b**) Comparison of the resistivity value of different types of TiO_2_ composites.

**Figure 8 sensors-22-05794-f008:**
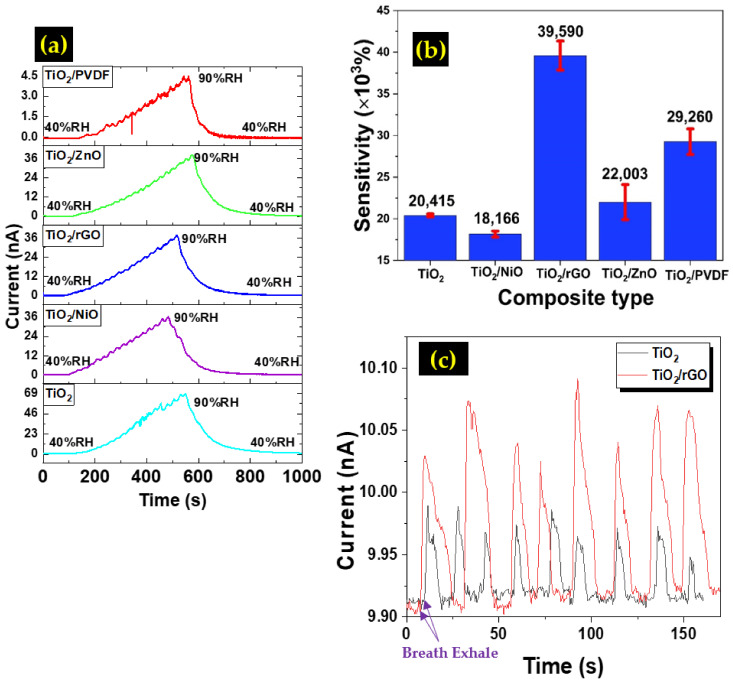
(**a**) Humidity response of the prepared humidity sensor, (**b**) Comparison of sensor response between different humidity sensors, (**c**) Breath test results for TiO_2_ and TiO_2_/rGO samples.

**Figure 9 sensors-22-05794-f009:**
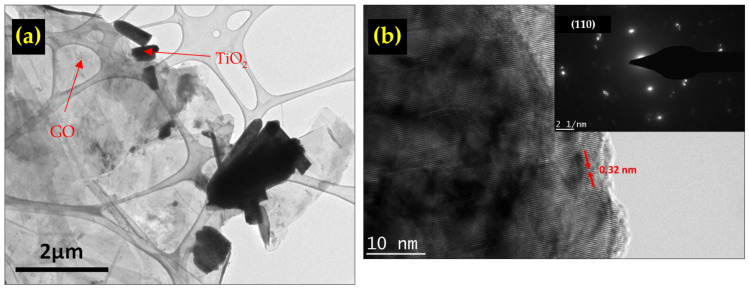
(**a**) TEM image of the TiO_2_/rGO composite, (**b**) SAED of TiO_2_/rGO.

**Figure 10 sensors-22-05794-f010:**
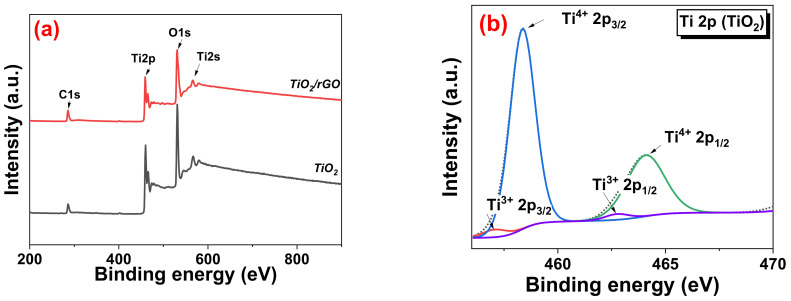
(**a**) XPS survey scan spectra, (**b**) narrow scan Ti 2p of TiO_2_, (**c**) narrow scan Ti 2p of TiO_2_/rGO, (**d**) narrow scan C 1s of TiO_2_/rGO, (**e**) narrow scan O 1s of TiO_2_, and (**f**) narrow scan O 1s of TiO_2_/rGO.

**Figure 11 sensors-22-05794-f011:**
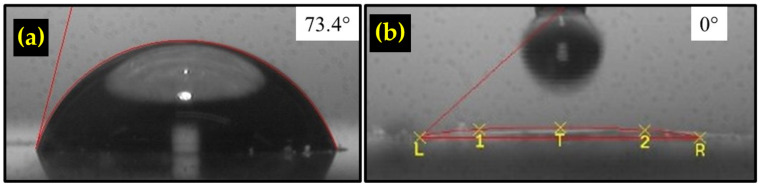
Contact angle measurement result of the (**a**) rGO thin film and (**b**) TiO_2_/rGO composite.

**Figure 12 sensors-22-05794-f012:**
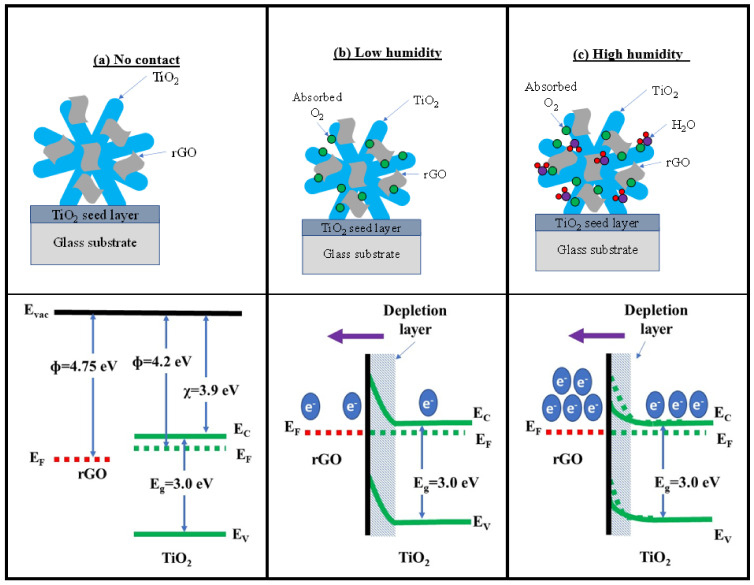
Proposed energy band diagram of the TiO_2_/rGO humidity sensor.

**Table 1 sensors-22-05794-t001:** Diffraction angle, interplanar spacing, lattice constant, crystallite size, microstrain, and dislocation density of different TiO_2_ composites.

Composite Type	Diffraction Angle, 2*θ* (º)	Interplanar Spacing, *d_hkl_* (Å)	Lattice Constant, a (Å)	Crystallite Size, D (nm)	Microstrain, ε (10^−3^)
TiO_2_	26.88	3.31	4.68	97.2	1.72
TiO_2_/NiO	27.16	3.32	4.64	75.4	1.86
TiO_2_/rGO	26.88	3.31	4.69	38.9	1.74
TiO_2_/ZnO	27.19	3.28	4.63	50.5	1.26
TiO_2_/PVDF	26.87	3.31	4.69	65.5	1.92

**Table 2 sensors-22-05794-t002:** Hall effect measurement of TiO_2_ composites.

Sample	Carrier Concentration (×10^14^ cm^−3^)	Carrier Mobility (×10^3^ cm^2^/(V·s))
TiO_2_	8.24	0.26
TiO_2_/NiO	0.79	1.55
TiO_2_/rGO	9.93	3.46
TiO_2_/ZnO	1.42	1.62
TiO_2_/PVDF	3.34	0.28

**Table 3 sensors-22-05794-t003:** Comparison of sensor response with other composite-based humidity sensors.

Material	Sensor Type	Humidity Range	Sensor Response [Calculation Formula]
MWCNT/Polyacrylic acid [48]	Resistive	30–90%RH	913.8% [$\left(\Delta R/R_{o}\right)\times 100\%$]
Graphene/Cellulose [49]	Resistive	5–90% RH	290% [$\left(\Delta R/R_{o}\right)\times 100\%$]
ZnO/SnO_2_ [50]	Resistive	40–90% RH	75,440% [$\left(R_{40}/R_{90}\right)\times 100\%$]
Mn-doped NiO/NiO [51]	Resistive	40–90% RH	27,000% [$\left(R_{40}/R_{90}\right)\times 100\%$]
MWCNT/hydroxyethyl cellulose [52,53]	Resistive	20–80% RH	290% [$\left(\Delta R/R_{o}\right)\times 100\%$]
Graphene/Methyl-red [54]	Resistive	5–95% RH	9636% [$\left(\Delta R/R_{o}\right)\times 100\%$]
Cellulose nanofiber/graphene nanoplatelet [55]	Resistive	30–90% RH	14,000% [$\left(\Delta R/R_{o}\right)\times 100\%$]
Cellulose nanofibers/CNT [56]	Resistive	11–95% RH	6990% [$\left(\Delta I/I_{o}\right)\times 100\%$]
TiO_2_/rGO (Present work)	Resistive	40–90% RH	39,590% [$\left(\Delta R/R_{o}\right)\times 100\%$]

## Data Availability

Data is contained within the article.

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
