# Peer review of "Evaluating Different TiO2 Nanoflower-Based Composites for Humidity Detection"

_sensors, 2022, doi:10.3390/s22155794_

Round 1

Reviewer 1 Report

The authors evaluated different TiO2 nanoflower-based composites for humidity detection. As complementary materials, they studied NiO, ZnO, PVDF and rGO. They used many complementary characterization techniques. They focused the study on the TiO2-rGO composite because they achieved the highest sensor response with it.

The paper is well written, the figures are accurate and clear, and there are many references that are recent. 

Nevertheless, I have some comments to improve the quality of the article which is good as it is.

In the introduction, line 46, you talk about the benefits of TiO2, including its non-toxicity. I disagree with you because the World Health Organisation's International Agency for Research on Cancer (IARC) has determined that titanium dioxide is a “possible carcinogen for humans”.

In Figure 3, you have specified the scale for the FESEM images. I suggest adding the scale for the inserts as well.

Some terms are not defined in equations (1), (2) and (3) which are not referenced. What is the difference between d and dhkl

In your reference [27], we can see: 1/d2 = 4 (h2 + hk + k2) / 3a2 + (l2/c2). This is different from your equation (2). How do you explain this difference?

In the same reference [27], we can see that a = 4.16 Å for TiO2 and you determine a = 4.68 Å (in Table 1). Could you add a comment regarding this difference?

In the same Table 1, you also give the values of D and ? for TiO2. They are also quite different from those found in your reference [28]. Could you add a comment regarding this difference?

Regarding Figure 5, I regret the lack of comment. Indeed, you only comment on the presence of the D and G bands due to the RGO. What is the point of calculating the ID/IG ratio? In Figure 5 (a), you don’t mention the B1g band. The same is true for the * band. Also, it is possible to determine whether it is anastase or rutile. 

In line 225 of the Figure 6 (b) legend, I think it is “Comparison of the resistivity…” not conductivity.

In your reference [33], we can see a carrier mobility value of 6.3x103 cm2/V.s for TiO2 with a lower carrier concentration value. This value is very different from yours. Could you add a comment on this point?

In line 249: ‘’ For the TiO2/NiO sample, there is a slight decrease in the sensor response value when compared with the undoped sample, which might be due to the increased resistivity of the nanocomposite. “ The resistivity of the TiO2-ZnO composite is higher than the resistivity of TiO2/NiO but the sensitivity is also higher. So, I don’t understand your argument.

In Table 3, you have compared the response of the sensor with other composite-based humidity sensors. This is interesting but I think the comparison is quite difficult. Indeed, the sensor response sometimes concerns a current or a capacitance or a resistance…It would be more interesting from my point of view to compare the sensor responses with the same units or with the same composite to highlight the performance of your sensor.

A sensor can be qualified by its response but also by its response time. Could you make a comparison of your composites using this aspect and also with other commercial or academic sensors?

In line 303: “The deconvolution of the spectra reveals 5 peaks that can be assigned to carbon from C=C sp2, C=C sp3, C-O, C=C, and O-C=O bonds.” But in Figure 9 (d), we can see: C=C sp2, C=C sp3, C-O, C=O, and O-C=C which is different.

In line 347: “As shown in Figure 8(a), the interconnection between rGO and the TiO2 nanorod structure would enable the efficient electron transfer between them.” I agree, but do you think that a TEM image can be representative of the whole sample?

In Figure 11, you use values for Eg? and ?. Could you reference these values?

Reviewer 2 Report

Dear authors, I appreciate for a nice article on nanoflowers. I suggest the following modifications to be done to improve the quality of the paper-

1. Figure 1 should be modified a little as per the abstract by including better sensing ability of rGO based NFs materials

2. FESEM images: It would be nice if EDS/EDX/mapping is also accompanied with SEM images for better justifications of the composite nanoflowers

3. Figure 4 b is not well explained in the text.

Round 2

Reviewer 2 Report

Manuscript is improved and can be accepted now.